# An Actuator Allocation Method for a Variable-Pitch Propeller System of Quadrotor-Based UAVs

**DOI:** 10.3390/s20195651

**Published:** 2020-10-02

**Authors:** Ching-Wei Chang, Shengyang Chen, Chih-Yung Wen, Boyang Li

**Affiliations:** 1Department of Mechanical Engineering, The Hong Kong Polytechnic University, Kowloon, Hong Kong; chingwei.chang@connect.polyu.hk (C.-W.C.); shengyang.chen@connect.polyu.hk (S.C.); chihyung.wen@polyu.edu.hk (C.-Y.W.); 2Interdisciplinary Division of Aeronautical and Aviation Engineering, The Hong Kong Polytechnic University, Kowloon, Hong Kong

**Keywords:** unmanned aerial vehicle (UAV), actuator allocation, variable-pitch propeller, flight experiment, Simulink simulation

## Abstract

This paper presents a control allocation method for enhancing the attitude following performance and the energy efficiency of a variable-pitch propeller (VPP) system on quadrotor-based unmanned aerial vehicles. The VPP system was modeled according to the blade element momentum (BEM) theory, and an actuator allocation method was developed with the aim of enhancing the attitude control and energy performance. A simulation environment was built to validate the VPP system by creating a thrust and moment database from the experiments. A four-motor variable-pitch quadrotor was built for verifying the proposed method. The control allocation method was firstly verified in a simulation environment, and was then implemented in a flight controller for indoor flight experiments. The simulation results show the proposed control allocation method greatly improves the yaw following performance. The experimental results demonstrate a difference in the energy consumption through various pitch angles, as well as a reduction in energy consumption, by applying this VPP system.

## 1. Introduction

With the emerging Microelectromechanical Systems (MEMS) technology, the weight, volume, and price of microsensors and actuators are significantly reducing [1]. The manufacturing and development of unmanned aerial vehicles (UAVs) both benefit from this trend of miniaturization [2,3]. Consequently, UAVs have started to gain attention for use in domestic applications due to their multifunctional air superiority and affordable price [4,5,6]. The performance requirements of UAV design are increasing rapidly, due to the expansion of a variety of UAV applications. Nowadays, end-users expect UAVs to have not only long endurance and stable flight performance, but also an extended flight range with increasing speed and enhanced maneuverability. Considering the demand for vertical take-off and landing (VTOL) with high flying speeds, traditional multirotor UAVs do not seem to be able to meet the above-mentioned requirements in many application scenarios.

Quadrotor-based UAVs include the commonly seen quadrotor and the VTOL type, both with four propulsion systems and wings. A series of novel designs for VTOL UAVs, such as tail-sitters and tilt-rotors, merge the quadrotor-type structure into a general fixed-wing design [7,8]. This combined layout provides the UAV with the ability to take-off vertically in a limited space, similarly to a quadrotor, and shifts into the horizontal flying mode after reaching a safe altitude, as does a fixed-wing UAV [9]. This feature provides the opportunity for the UAV to enhance its aerodynamic efficiency and operability in a populous and highly urbanized region, such as Hong Kong. However, the combination of fixed- and rotary-wing structures inevitably causes some issues, such as control instabilities.

Generally, quadrotor-based VTOL UAVs use the same propulsion system in both vertical and horizontal flying modes; however, the energy efficiency of the propulsion system needs to be addressed. The large flight envelope that this type of UAV experiences constitutes a great challenge in the design of its propulsion system, as it requires the generation of precise thrust across a large range of airspeed. Another common problem of quadrotor-based VTOL UAVs is their large wing area, which may encounter a significant crosswind and thus generate an enormous force on the fuselage of the UAV, making it difficult to maintain the vehicle’s position during a maneuver in the vertical stage [10]. Generally, the control strategy applied to the vertical phase of quadrotor-based VTOL UAVs, especially tail-sitters, is similar to that of typical quadrotor UAVs [11], where the roll and pitch moments are generated by means of re-distributing the output thrust of the propellers, while the yaw moment results from the differential moments generated by the propellers. Through multiplying the large forces generated by the propellers with the corresponding moment arms, the pitch and roll moments are produced, which is much easier than generating the yaw moment through the differential moments or propellers. Hence, quadrotor-based VTOL UAVs such as tail-sitters with large wings have greater difficulty in maintaining their yaw stability when facing a crosswind.

Variable-pitch propeller (VPP) systems, which are broadly used in manned aircrafts, were originally designed to generate varied thrust and to optimize the energy efficiency of combustion engines [12]. The main idea of a VPP system is to change the angle of attack (AOA) of propellers for altering the generated thrust, along with maintaining a reasonable aerodynamic efficiency across different inflow airspeeds [13]. Nonetheless, while changing the AOA of a VPP system, the generated moment of a single VPP system also varies. Thus, VPP systems are a potential solution for overcoming the shortcomings of the quadrotor-based UAV systems.

VPP systems have already been implemented by researchers in various kinds of small UAV systems. For easy classification, they are sorted by the number of main propellers; those using a single set of main propellers are more akin to traditional helicopters. The cyclic pitch is engaged for generating moments in the roll and pitch axes to maintain their attitude in the hovering mode. As for the yaw axis, smaller auxiliary propellers are used only for generating the thrust to counter a large amount of moment from the main rotor [14,15]. The twin-rotor VPP VTOL design uses the same control strategy on the pitch and roll axes. but counters the yaw moment by using two counter-rotating main propellers [16]. The four-rotor-type VTOL demonstrated by Chipade et al. [17] greater resembles a quadrotor, which uses a single powerplant to drive the four propellers at a constant speed and is controlled by changing the AOA of each propeller only.

VPP systems are also widely used in typical quadrotor UAVs, which can be divided into two categories by the number of powerplants. The first is those with a single powerplant driving four propellers with a belt or a transmission shaft. This setup, combined with an internal combustion engine with a high energy density, can guarantee the long endurance of the UAV [18,19]. Some other works with a single direct current (DC) motor also seek the optimal energy efficiency by varying the AOA at a single rotational speed for the four propellers [20]. The second category uses an independent electric motor on every VPP unit. These categories with DC motor powerplants take advantage of the performance of the actuator, in which servo motors respond faster than increasing the rotational speed of the propellers, achieving agile maneuvers such as flipping upside down [21,22,23]. To enhance the performance of small fixed-wing UAVs, the use of a VPP system has been considered. To control their attitude with less deformation on the wings, morphing UAVs [24] were equipped with a VPP system acting as the propulsion system, and the cyclic pitch was used to generate control moments. Both Henderson and Papanikolopoulos [25] and Manchin et al. [26] used the VPP system to maximize the aerodynamic efficiency of the propellers by varying the AOA according to the detected airspeed.

Despite the previous works regarding the application of VPP systems to small UAVs, no efforts have been made to increase the performance of the yaw stability and energy consumption of quadrotor-based UAVs. Due to the dual-signal input characteristic of the VPP system, a single actuator set gains the ability to provide more than a single combination of thrust and moment. This characteristic consequently provides a broader range of outputted force and moment, yielding a better flight performance of the whole UAV system. However, the added actuators and the over-actuated characteristic of the VPP system bring the complexity to the controller and actuator allocation comparing to other small UAVs. In particular, the yaw moment generated by VPP systems has not been emphasized in previous works. Stability in the yaw axis, which is crucial during the take-off and landing stages of a quadrotor-based VTOL UAV, influences the initial state of the transition stage of the aircraft and affects the safety of transition into the horizontal flying mode. VPP systems can also generate a larger amount of moment to provide a considerably greater control margin for UAVs than can fixed-pitch propulsion systems.

The main contribution of this work is to introduce an effective control allocation method for implementing a VPP system into a quadrotor-based UAV. The proposed approach aims to improve the energy efficiency, attitude and position control of all kinds of quadrotor-based UAV systems during hover flight. A simulation testbed is built with data sets collected from static thrust experiments for validating and testing the proposed method. At last, flight tests are also accomplished with self-designed VPP systems mounted on a small quadrotor-type UAV.

The rest of this paper is organized as follows. Section 2 describes the modeling, allocation method, and a static thrust test for the VPP system. Section 3 introduces the Simulink simulation environment, followed by the simulated results. The flight tests and results are then presented in Section 4. At last, conclusions are presented in Section 5.

## 2. VPP Modeling and Control Allocation

### 2.1. VPP Modeling

The modeling of the VPP system was inspired by both traditional helicopter rotors and small multi-rotor UAV systems. The lift and drag generated by the system were evaluated separately using the blade element momentum (BEM) theory. In this section, the numeric modeling of the VPP system is described. Considering the general setup of a quadrotor-based UAV, which does not have an overlapping section on its blade surface, complete analysis of the whole UAV was not necessary. Hence, the following modeling process considers each VPP unit individually.

In this proposed UAV system, the incoming airflow toward the system is relatively slow. As a result, air is assumed to be incompressible and inviscid, and the inflow angle for the propeller blade of the system is considered very close to 0°. All stall conditions are ignored in the evaluation process.

Figure 1a shows a rotating disk of a VPP system. The blade was broken down into several small pieces in the spanwise direction for analysis. Suppose dr is the width of the blade element, r the distance to the rotating center, and c the chord length of the blade element. With these variables, the generated lift of the blade element can be written as:(1)$$dL=\frac{1}{2}\;\rho \;u^{2}\;C_{l}\;c\;dr,$$
where the velocity u is the airflow speed through the airfoil element, ρ the density of air, and $C_{l}$ the lift coefficient of the airfoil, which is proportional to the AOA α of a certain blade element. $C_{l}$ is then determined with a slope of lift coefficient $k_{l}$ as $C_{l}=k_{l}\;\alpha$.

The more specific airfoil geometry, velocity, and forces are defined in Figure 1b. The propeller’s forward speed u can be calculated by multiplying the propeller’s rotational speed Ω with the distance to the propeller’s center r. Table 1 summarizes all of the symbols used in Figure 1b.

The aerodynamic lift generated by the propeller can then be rewritten as:(2)$$dL=\left[\frac{1}{2}\;\rho \;{\left(\Omega \;r\right)}^{2}\right]\;\left(k_{l}\;\alpha \right)\;c\;dr.$$

By integrating the lift along the whole length of the rotor blade R, the total lift generated by the blade is expressed as:(3)$$L=\frac{1}{6}\;\rho \;\Omega^{2}R^{3}\;k_{l}\;c\;\alpha .$$

The total thrust generated by the system is then determined by multiplying Equation (3) by the total number of blades n. In this specific system, the chord length c, blade length R, and airfoil coefficient remain constant. Thus, the total thrust of a propeller becomes:(4)$$T=K_{T}\;\Omega^{2}\;\alpha ,$$
where $K_{T}$ is known as the thrust coefficient,
(5)$$K_{T}=\frac{n}{6}\;\rho \;R^{3}\;k_{l}\;c.$$

After evaluating the lift of the propeller, the torque generated by the propeller is also calculated. Drag needs to be considered in two parts, namely lift-induced drag and aerodynamic drag. The aerodynamic drag is the part of force generated by the airfoil:(6)$$dD_{o}=\frac{1}{2}\;\rho \;{\left(\Omega \;r\right)}^{2}\;C_{d}\;c\;dr,$$
where $C_{d}$ is the drag coefficient of the airfoil. Similar to the lift coefficient, the drag coefficient is also treated as proportional to the AOA before the airfoil stalls, i.e., $C_{d}=k_{d}\;\alpha$, where $k_{d}$ is the slope of the drag coefficient.

For small UAVs, the airspeed toward the rotational plane (V) is relatively low (when hovering, it is considered 0); therefore, the inflow angle ϕ is considered very small. The small-angle approximation is applied in this situation, and the lift-induced drag is expressed as:(7)$$dD_{i}=dL\;\phi .$$

The total drag is then calculated by combining these two types of drag together. The drag generated by a piece of blade element is:(8)$$dD=\left[\frac{1}{2}\;\rho {\left(\Omega \;r\right)}^{2}\right]\;(C_{l}\;\phi +C_{d})\;c\;dr.$$

After multiplying Equation (8) with *r* and integrating it through the whole blade, the torque induced by the blade drag Q and its torque coefficient $K_{Q}$ are derived as:(9)$$\begin{array}{c} Q=K_{Q}\;\Omega^{2}\;\alpha ,\\ K_{Q}=\frac{n}{8}\;\rho \;R^{4}\;c\;\left(k_{l}\;\phi +\;k_{d}\right). \end{array}$$

### 2.2. VPP Control Allocation

The coordinate system used for our modeling is shown in Figure 2. The world coordinate system ($\Gamma^{w}$) is defined with the fixed North–East–Down (NED) frame, and the vehicle’s body coordinate system ($\Gamma^{b}$) is located at the center of mass of the UAV system with the $X^{b}$ axis pointing forward and the $Z^{b}$ axis pointing downs. Propellers 1 and 2 rotate clockwise, while propellers 3 and 4 rotate counterclockwise. 

The position and attitude controllers of this work were adopted from the PX4 flight stack [27]. The output commands from the attitude controller are three moments and a force, which can be described as $\tau_{x},\;\tau_{y},\;\tau_{z},\;\mathrm{and}\;T_{total}$. The moments are all defined according to the UAV’s body frame, and the thrust points in the negative direction of the $Z^{b}$ axis.

Let the thrust generated by each motor be $T_{i}$ and the moment be $Q_{i}$. Then, the moments and thrust according to the UAV frame are written as:(10)$$\left[\begin{array}{c} \tau_{x}\\ \tau_{y}\\ \tau_{z}\\ T_{total} \end{array}\right]=\left[\begin{array}{c} -T_{1}d+T_{2}d-T_{3}d+T_{4}d\\ T_{1}d-T_{2}d+T_{3}d-T_{4}d\\ Q_{1}+Q_{2}+Q_{3}+Q_{4}\\ T_{1}+T_{2}+T_{3}+T_{4} \end{array}\right],$$
where d is the distance of each motor to the center of gravity. By combining Equation (10) with the VPP model (Equations (4) and (9)) and a motor model $\left(M_{i}=\Omega_{i}^{2}\;\alpha_{i}\right)$, it can be concluded as:(11)$$\left[\begin{array}{c} M_{1}\\ M_{2}\\ M_{3}\\ M_{4} \end{array}\right]={\left[\begin{array}{cccc} -K_{T}d & K_{T}d & -K_{T}d & K_{T}d\\ K_{T}d & -K_{T}d & K_{T}d & -K_{T}d\\ K_{Q} & K_{Q} & -K_{Q} & -K_{Q}\\ K_{T} & K_{T} & K_{T} & K_{T} \end{array}\right]}^{-1}\left[\begin{array}{c} \tau_{x}\\ \tau_{y}\\ \tau_{z}\\ T_{total} \end{array}\right].$$

In this moment, the relationship between the four control outputs and the eight actuator setpoints ($\Omega_{i}\;$ and $\;\alpha_{i}$, *i* = 1–4) is connected. With four control outputs but a total of eight actuator setpoints, the over-actuated system provides a certain amount of redundancy for the flight controller. The objective of the proposed allocation method is to increase the attitude tracking performance (especially in the yaw axis) of quadrotor-based UAVs, together with minimizing the energy consumption.

The minimization of energy consumption is achieved by meeting the thrust demand for hovering and putting the result into the experimental VPP model, which is described in the following section, to find out the optimal initial AOA ($\alpha_{init}$) for the system. After retrieving $\alpha_{init}$, four gains ($G_{x},\;G_{y},\;G_{z},\;\mathrm{and}\;G_{T}$) are introduced, determined to enhance the UAVs’ performance in roll, pitch, yaw, and altitude, respectively. The AOAs on each VPP unit can then be determined as:(12)$$\left[\begin{array}{c} \alpha_{1}\\ \alpha_{2}\\ \alpha_{3}\\ \alpha_{4} \end{array}\right]=\left[\begin{array}{c} \alpha_{init}\\ \alpha_{init}\\ \alpha_{init}\\ \alpha_{init} \end{array}\right]+\left[\begin{array}{cccc} -1 & 1 & 1 & 1\\ 1 & -1 & 1 & 1\\ -1 & 1 & -1 & 1\\ 1 & -1 & -1 & 1 \end{array}\right]\left[\begin{array}{c} G_{x}\lambda_{x}\\ G_{y}\lambda_{y}\\ G_{z}\lambda_{z}\\ G_{T}\lambda_{T} \end{array}\right],$$
where $\lambda_{x},\;\;\;\lambda_{y},\;\;\mathrm{and}\;\;\lambda_{z}$ are the moment commands generated from the attitude controller, and $\lambda_{T}$ is the incremental thrust command after removing the hovering thrust from the total thrust ($\lambda_{T}=T_{total}-T_{hover})$.

After checking that the calculated AOAs did not exceed the physical boundary of the VPP system, the AOAs can be used for determining the rotation speed of the system by:(13)$$\Omega_{i}=\sqrt{\frac{M_{i}}{\alpha_{i}}}.$$

### 2.3. Static Thrust Experiment

A static thrust experiment of the VPP system is introduced in this section. The setup of the experimental hardware is shown in Figure 3. The whole system was mounted directly on an ATI Mini40 F/T sensor (ATI Industrial Automation, Apex, NC, USA), which was fixed on a metal tube assembled on the workbench. The whole system contained two separate actuators. The first one was a digital servo Emax ES09MD (EMAX model, Shenzhen, China) to control the pitch angle and the second was a motor and its electronic speed controller (ESC). The motor was the Sunnysky X2814 KV1250 (Sunnysky, Zhuhai, China), and the ESC was a Platinum 50A V3 from Hobbywing (Hobbywing, Shenzhen, China). The rotation speed and the pitch angle were controlled by the Arduino Mega, which was programmed with a built-in closed-loop governor.

The force, torque, and electric current data were measured at various rotational speeds and pitch angles. The electric power source in this experiment was the Chroma 62012p DC power supply (Chroma ATE Inc., Taoyuan, Taiwan), which was set at a fixed voltage (12.6 V). During each measurement, a fixed rotational speed (RPM) was set and the pitch angle was changed and maintained for 20 s. Force and torque data were collected through a NI-9220 DAQ system (National Instruments, Austin, TX, USA), and the power consumption information was provided directly from the DC power supply. Finally, the actuators’ control signals and rotational speed during the test period were collected through the Arduino.

Figure 4 shows the results of the static thrust experiment. The response surfaces of the thrust, moment of force, energy consumption, and efficiency to the pitch angle and rotational speed of the propeller are plotted. The thrust per unit watt is defined as the thrust divided by the power used by the motor ($E_{T/P}=Thrust/Power$). According to the experiment results, the pitch angle of the maximum energy efficiency at zero airspeed was approximately α=12°, which was set to be the $\alpha_{init}$ of the proposed system to minimize the power consumption under hover conditions.

## 3. Simulation

A simulation testbed was set up for verifying and testing the proposed allocation algorithm. The proposed method was applied to several test scenarios to compare its performance with that of a traditional fixed-pitch propulsion system.

### 3.1. Simulation Environment

The simulation environment was built up within the MATLAB Simulink environment. The built-in Aerospace Blockset was used to represent the equations of motion of the UAV system. Figure 5 represents the workflow of the simulation. The flight control strategy of this study was adopted from the PX4 flight stack, which is a series of cascade proportional–integral–derivative controller (PID) control structures. The control system contained three sections: Position controller, attitude controller, and control allocation system. 

In the simulation environment, an X-type quadrotor was used to demonstrate the proposed allocation method. The quadrotor was set to use the same propulsion system from the static thrust experiment, with an arm length of 25 cm and the total weight of 2 kg.

The simulation environment can be pre-scheduled with a series of commands, which are the desired three-dimensional position setpoints ($position_{sp}$) and a heading setpoint ($\psi_{sp}$). When the position controller receives the preplanned commands, it calculates the position and velocity errors from the feedback references of the UAV dynamic model and further generates the desired acceleration using the PID controller. The desired acceleration setpoints were further calculated into attitude commands and a thrust command, which are shown as $\phi_{sp}$*,*$\;\theta_{sp}$, $\psi_{sp}$*,* and $T_{sp}$ in Figure 5, allowing the mono-thrust vector configuration vehicle to track its desired attitude. After the position controller, the attitude controller plays its part by generating force and moments with the feedback of the attitude and angular velocities. At the end of the flight control stage, the controller’s output is sent to the control allocation node. In this node, the four control outputs use the proposed control allocation method, translating the moments and force into four ESC signals and four servo signals. Finally, the actuators’ command signals are sent into the actuator dynamic model and then the UAV dynamic model.

As shown in Figure 6, the UAV dynamics model was constructed by two look-up tables and a six-degrees-of-freedom (6DOF) equation of motion (EoM) model. Both the ESC and servo control signals went through a set of motor dynamics look-up tables, generating a rotation speed signal according to different input strengths and aerodynamic drags. The pitch servo control signal and the rotation speed were then fed into the variable-pitch dynamics model. After the look-up table, the forces and moments of four sets of independent propulsion units were produced. As for the last 6DOF EoM block, all of the generated forces and moments were gathered, after which, the 6DOF EoM model then figured out the corresponding positions, velocities, attitudes, and angular velocities. This generated information was moved on to the controller for iteration.

### 3.2. Simulation Results

In this section, the proposed variable-pitch allocation method is validated and compared to a general fixed-pitch propeller system in the simulation testbed. Three flight scenarios were tested; the first scenario was position holding with a yaw step, followed by position tracking with a step command, while the third was waypoint mission tracking. The initial conditions and disturbances of all simulations were set equally. In the following results, the setpoints are shown in dashed lines, and the feedback signals from the UAV dynamics model are shown in blue and red lines, where blue lines represent the results of the system without VPP, and the red ones are with VPP activated.

#### 3.2.1. First Scenario: Position Holding with a Yaw Step Command

Figure 7 shows a comparison of the two different actuator systems with and without VPP responding to an aggressive yaw command. The command and respond yaw signal (ψ) are shown on the bottom right-hand side. The yaw command was given at t = 10 s with an amplitude of 0.4 rad (23 degrees). Comparing the two actuator systems, the proposed VPP system provided a 100 ms faster response time. The overshoot of the proposed VPP system was also smaller than that of the general fixed-pitch system. Due to the operation of AOA, the subsequence effects on the position (in the left column) diverged in the two cases. When applying the VPP controller, the altitude (Z axis position) climbed 0.65 m, but the altitude in the fixed-pitch data set dropped 0.45 m. In both the X and Y- axes, the positions diverged in the two test sets, but the pitch (θ) and roll (ϕ) attitude remained very similar to one another, indicating that the VPP controller did not have a large influence on the other axes despite improving the yaw performance.

#### 3.2.2. Second Scenario: Position Tracking with a Step Command

Figure 8 presents the simulation results of the tracking performance of the systems with and without VPP under a step position command. The command and response position of the X axis are shown in the top left, in which a 1.5 m step position command was generated at t = 10 s. Similar to the results of the previous position holding simulation, the VPP system showed its strength in reducing the response time by approximately 100 ms and had a slightly smaller overshoot (5%) to that of the fixed-pitch system. In addition, while generating larger force to achieve a faster maneuver, the interferences against the altitude and heading (yaw) generally remained smaller for the VPP system.

#### 3.2.3. Third Scenario: Waypoint Mission Tracking

The last simulation scenario is displayed in Figure 9, which demonstrates a simple waypoint mission in the horizontal direction. The speed was set at ±0.1 m/s at each axis. In the left column, which shows the positions of the simulation, the following of the X, Y, and Z axes performed evenly during the gentle position commands. However, the main difference is the improvement in the yaw (ψ) direction. As shown on the bottom right-hand side of Figure 9, during the position movement, the yaw response with VPP activated was more stable compared to that without a VPP controller. 

#### 3.2.4. Simulation Summary

According to the simulation results, the VPP controller performed faster in both the yaw and step position signal following. As for the constant speed waypoints mission, the VPP system did not show advantages, since the triggering control input was relatively small. However, after calculating the root-mean-square error (RMSE) of the yaw direction for the three simulations, the RMSE results of these three simulation scenarios are shown in Table 2. Comparing them group by group, in the waypoint mission tracking set, applying the VPP controller reduced 79% of its RMSE. In the position step test set, the VPP controller also had a 58% reduction in its RMSE. Not to mention, even with a large yaw command, the yaw step data sets also show a 7% decrease in RMSE after applying the VPP controller. The RMSE results clearly demonstrate that the VPP system considerably improved the yaw control.

## 4. Flight Tests

Following the simulations, a variable-pitch quadrotor UAV was built. The detailed configuration of the UAV platform is illustrated below. The design of the quadrotor with the VPP system is introduced first, followed by the avionics and test environment. Finally, the flight test results are presented.

### 4.1. Flight Test Equipment

#### 4.1.1. UAV Design

As shown in Figure 10a, a typical X-type quadrotor structure was chosen for the test UAV. The diameter of the selected frame was 500 mm. At the end of each arm, a set of variable-pitch propulsion units were assembled. Figure 10b shows the Computer Aided Design (CAD) drawing of the proposed system, which is a combination of a brushless direct current motor (Sunnysky X2814 KV1250), an electronic speed controller (Hobbywing XRotor 40A), a set of variable-pitch propellers, and a hobby-grade digital servo motor (Emax ES08MD). The variable-pitch propellers were directly driven by the motor, and the variable-pitch mechanism used a four-bar linkage controlled by the servo motor. The pitch angle varied from +25° to −20°. The chosen propeller had a diameter of 28 cm, with NACA-0009 (National Advisory Committee for Aeronautics, Washington, DC, USA) airfoil with an 18-mm chord.

#### 4.1.2. Avionics and Test Environment

The avionics of the UAV system included a flight controller and an on-board computer. Pixhawk 2 (Hex Cube) (Hex Technology Limited., Hong Kong, China) with open-source PX4 firmware (v1.11.0dev) [27] was used as the basic flight control unit of the UAV. The installed firmware of the flight controller was a custom-built version that was programmed with the proposed control allocation method, commanding a total of four servo motors and four electronic speed controllers. The on-board companion computer was a UP Board [28] (AAEON Technology Inc., Taipei, Taiwan), which is a powerful companion computer integrated with an Intel® ATOM™ x5- Z8350 Processor (Intel Corporation, Santa Clara, CA, USA) and 4 GB DDR3L RAM that runs with Linux Ubuntu 18.04 and is installed with an ROS Melodic Morenia [29] distribution.

The flight experiments were conducted in an indoor laboratory equipped with the VICON (Vicon Motion Systems Ltd., Yarnton, Oxford, UK) motion capture system. The VICON motion capture system provides a real-time position and attitude information through the Wi-Fi signal to the avionics. The on-board computer was the communication center of the whole experimental setup. It communicated with the ground control station and the VICON system through the ROS network using the Wi-Fi and passed the information using the MAVROS [30] node to the flight controller. 

Preplanned waypoints and attitude commands were programmed as ROS nodes on the on-board computer. The tracking commands were generated and input into the flight controller in a fixed schedule, providing a controllable quantitative result for the flight tests. The network structure is shown in Figure 11.

### 4.2. Flight Test Results

This section outlines the flight experiments that were carried out to test the proposed method in a realistic flight environment. Similar scenarios to that in the simulation section were adopted in the flight tests, including the three scenarios: (1) Step yaw command, (2) step position command, and (3) waypoint mission tracking. Furthermore, power consumption during each flight was recorded for comparison.

#### 4.2.1. First Flight Test Scenario: Position Holding with a Step Yaw Command

In the first flight test, the aggressive yaw command scenario (90° to−90°) was tested. As shown in Figure 12, while the UAV was chasing the yaw command (the bottom right figure), whether applying the VPP controller or not, the yaw following speed remained even. However, a subsequent effect can be observed on the pitch (θ) and roll (ϕ) axes in terms of a slight reduction in their peak value. As can be seen in the bottom left figure, which shows the Z position (altitude) holding performance, the VPP controller also prevented large height changes during the yaw step.

Figure 13 shows the instantaneous power log during the position holding with a yaw step. As can be seen, at the moment of imposing the step yaw command, the power consumption of the two tests (with and without VPP) all rose significantly, but that using the VPP controller shows a smaller peak. In the whole test period, the power consumption of the UAV with the VPP system was lower than that without the VPP system. This is because the VPP controller used servos that consume less power to generate the same amount of moment.

#### 4.2.2. Second Flight Test Scenario: Position Tracking with a Step Command

In the second test (shown in Figure 14), a step position command of 1.5 m was adopted in the Y axis. Similar to the simulation results, the responses time of the VPP controller was slightly faster than that of the fixed-pitch set. The altitude and yaw during the fast position maneuver were also less affected for the VPP controller. 

Figure 15 presents the output signals from the controller to each actuator. The format of the signals sending to the actuators are in the radio control pulse-width modulation (PWM) standard, which has a minimum and maximum value of 1000 and 2000 μs. As shown, the motor signals have less shift when applying the VPP controller. Since motors consume significantly more power than servos, this provides the evidence of less energy cost for the VPP controller. The instantaneous power during the test is shown in Figure 16, which indicates less power consumption when applying the VPP controller under a similar position tracking performance directly.

The total energy usage in the above step yaw and position tracking tests is shown in Table 3. The total duration of the yaw test was 130 s, with a total of eight step commands, while in the step position tracking test, the duration was 110 s with a total of seven steps. In both test scenarios, using the VPP controller required less energy.

#### 4.2.3. Third Flight Test Scenario: Waypoint Mission Tracking

The last test scenario involved testing the power and energy consumptions at different initial pitch angles $\alpha_{init}$ through waypoint mission tracking. The waypoint mission was to fly along with a squared waypoint with the side length of 2 m twice. The vehicle kept a constant speed of 1.5 m/s. The three $\alpha_{init}$ were set at 12°, 14°, and 16°, respectively. From Figure 17, it can be seen that the tracking performance remained similar using different initial AOAs. The instantaneous power was lower for the UAV operated at the proposed $\alpha_{init}=12$° (Figure 18). The total energy consumption at different $\alpha_{init}$ is also compared in Table 4. Again, the initial setting at $\alpha_{init}=12$° outperformed the other two AOA angles.

## 5. Conclusions 

In this work, the objective was to increase the attitude following performance and reduce the energy consumption of a quadrotor-based UAV by implementing the VPP system. During this research, the VPP model was firstly evaluated using the BEM theory, followed by the introduction of the VPP actuator allocation method. The actuator allocation method was validated and tested in the simulation testbed and further implemented in the Pixhawk flight controller for indoor flight tests. From the results of both the simulation and flight tests, the allocation method was shown to be effective and can withstand high-intensity step maneuvers. Furthermore, the proposed method demonstrates an enhancement in the UAV performance of the yaw control with less effect on its altitude. The flight test results also showed energy consumption reduction by both varying the initial pitch angle and enabling the VPP controller during hovering status of the quadrotor-based UAV.

To obtain further improvements in the VPP system, attention will be focused on its application to a VTOL UAV system such as a tail-sitter. Wind tunnel experiments will be carried out to simulate the level flight cruising stage of the UAV. Then, a global energy optimization method for variable-pitch quadrotor-based VTOL aircrafts can be developed.

## Figures and Tables

**Figure 1 sensors-20-05651-f001:**
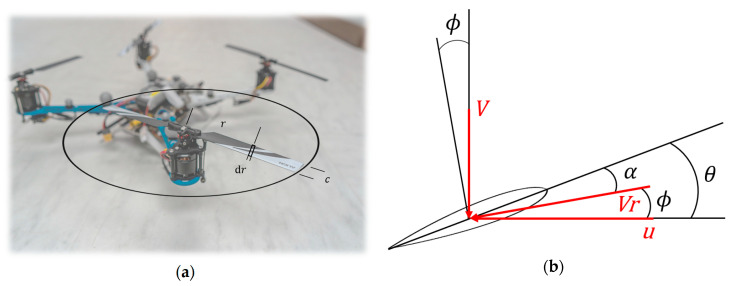
(**a**) Rotation disk of a variable-pitch propeller (VPP) system; (**b**) diagram to explain the blade element momentum (BEM) theory.

**Figure 2 sensors-20-05651-f002:**
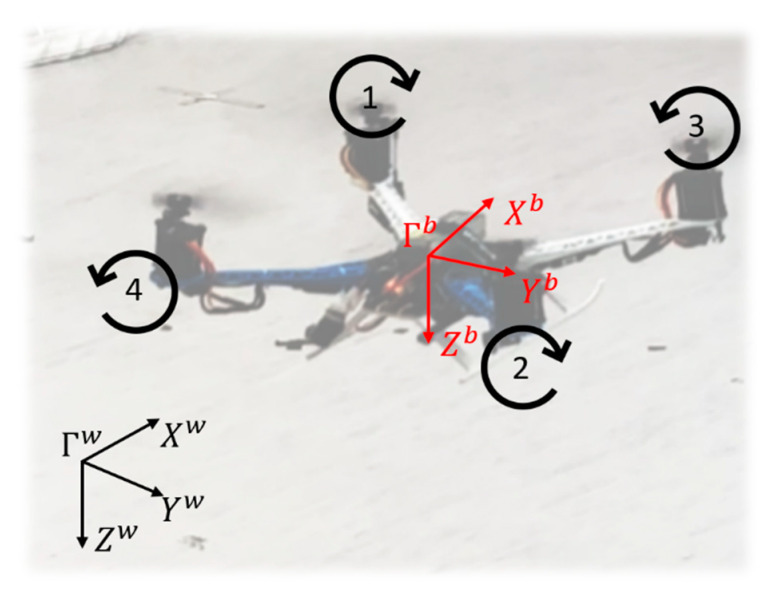
Coordinate system.

**Figure 3 sensors-20-05651-f003:**
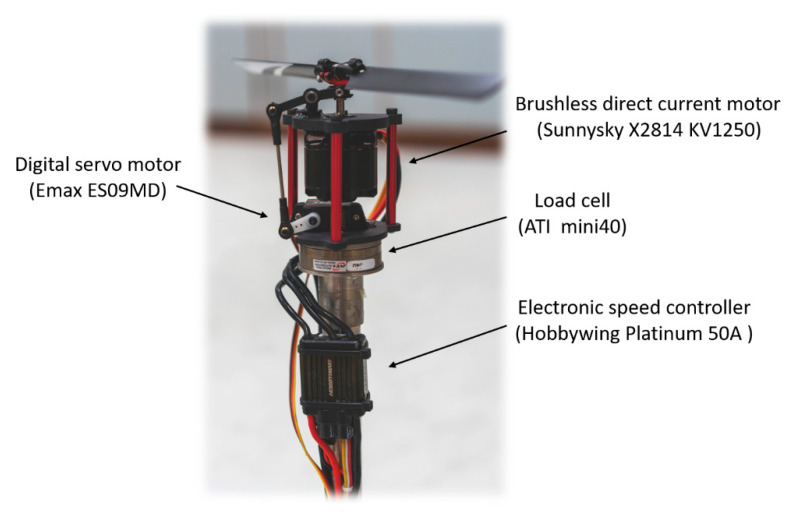
The set-up of the static thrust experiments.

**Figure 4 sensors-20-05651-f004:**
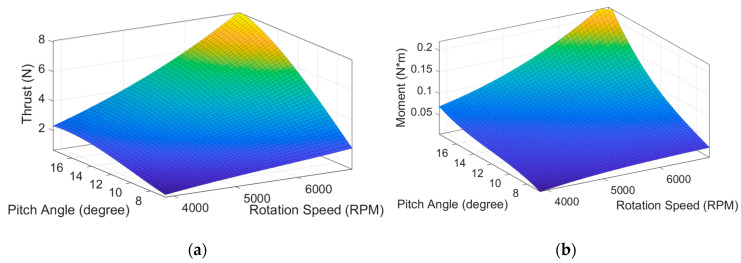
The experimental data of the VPP system, showing the response surfaces of the (**a**) thrust, (**b**) moment of force, (**c**) power consumption, and (**d**) thrust per unit watt to the pitch angle and rotational speed of the propeller.

**Figure 5 sensors-20-05651-f005:**
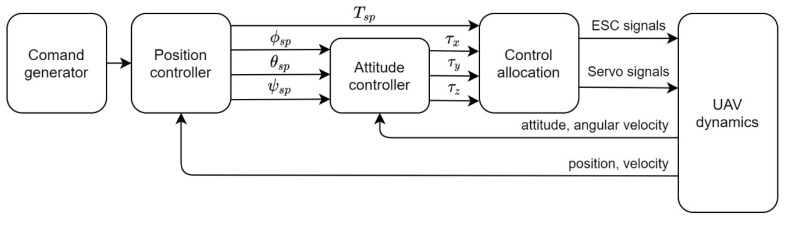
Flow chart of the simulation environment.

**Figure 6 sensors-20-05651-f006:**
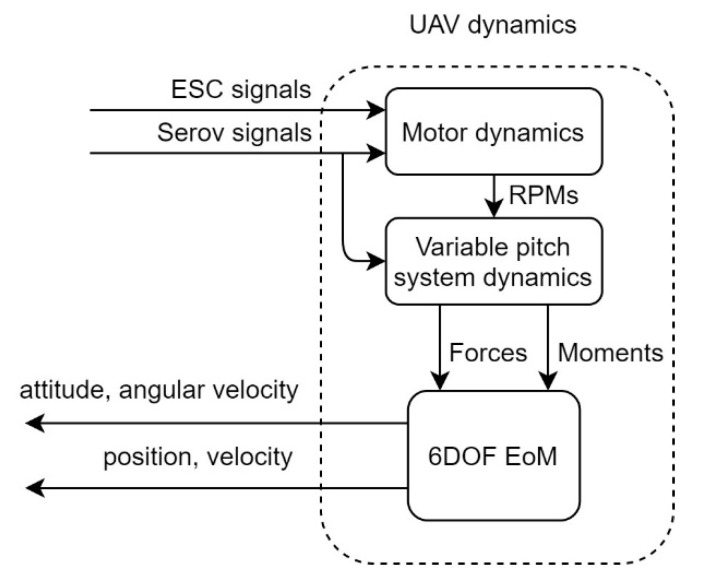
Flow chart of the unmanned aerial vehicle (UAV) dynamics model.

**Figure 7 sensors-20-05651-f007:**
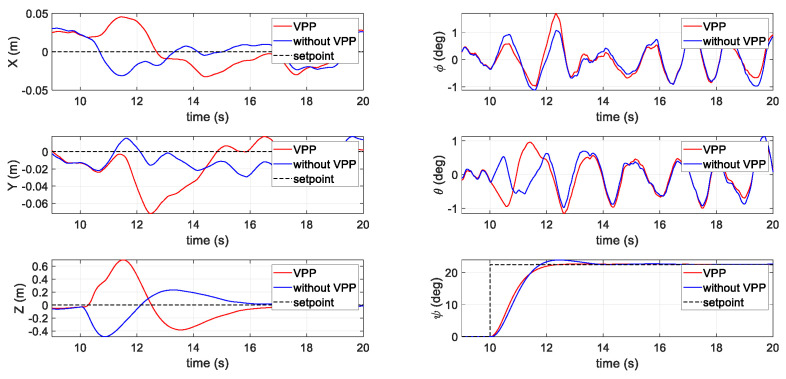
Results of the first scenario with a yaw step command in the simulation.

**Figure 8 sensors-20-05651-f008:**
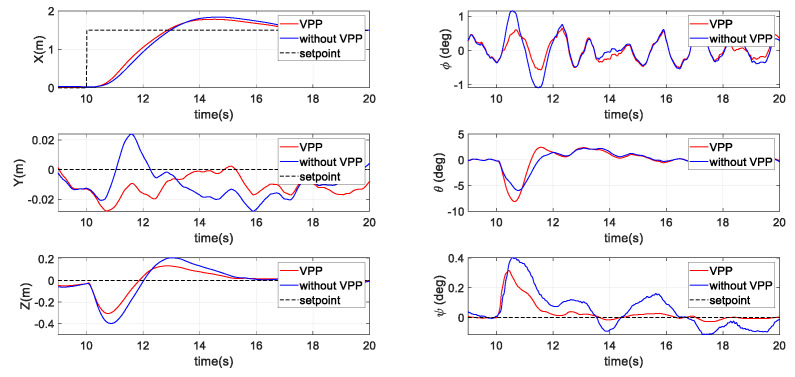
Results of the second scenario with a step command in the simulation.

**Figure 9 sensors-20-05651-f009:**
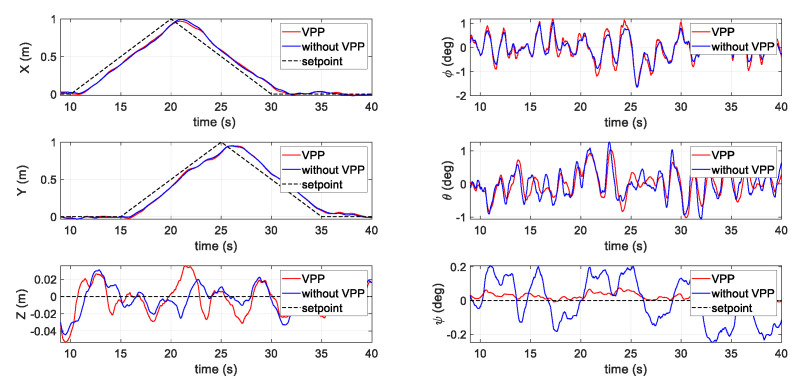
Results of the third scenario in the simulation.

**Figure 10 sensors-20-05651-f010:**
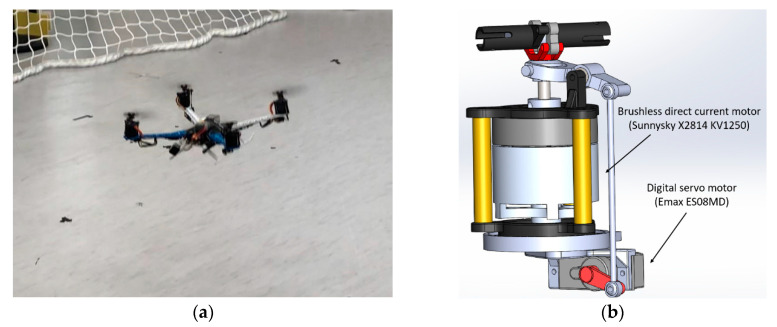
(**a**) Quadrotor flight test platform with the VPP system; (**b**) CAD drawing of the VPP mechanism.

**Figure 11 sensors-20-05651-f011:**
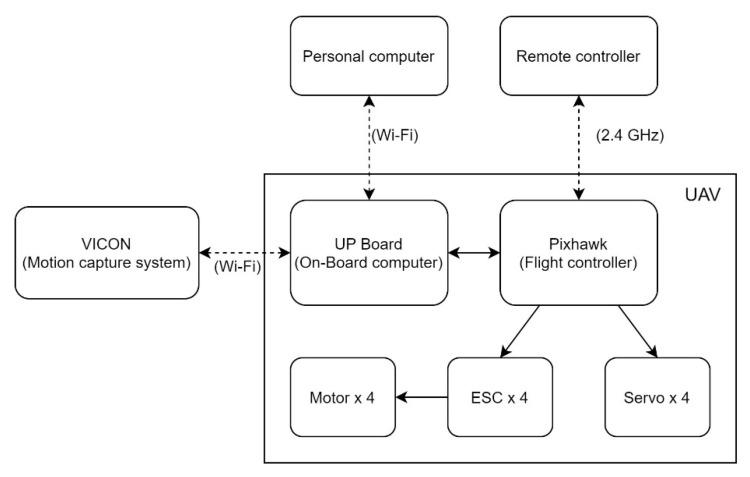
Avionics network structure.

**Figure 12 sensors-20-05651-f012:**
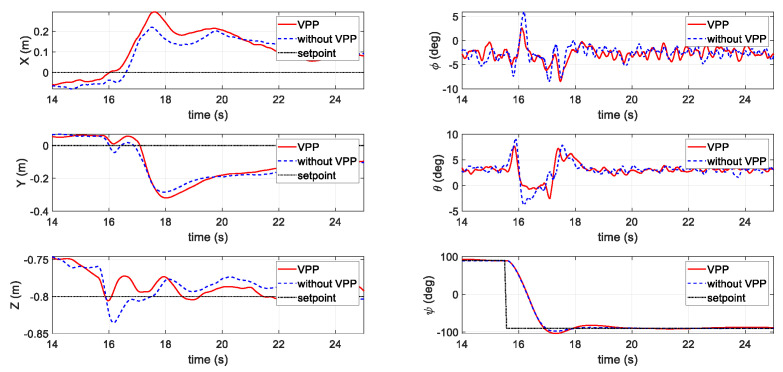
Results of the first flight test scenario with a step yaw command.

**Figure 13 sensors-20-05651-f013:**
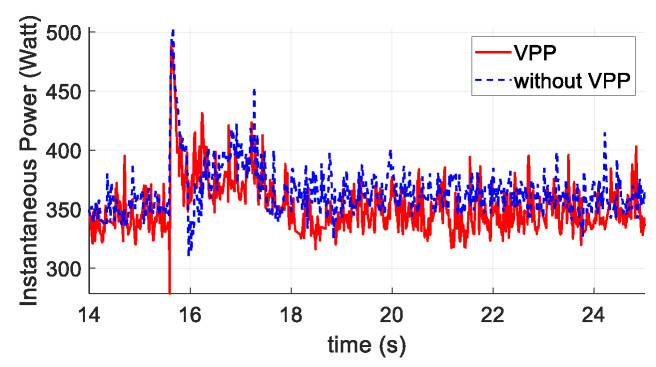
Instantaneous power during the first flight test scenario.

**Figure 14 sensors-20-05651-f014:**
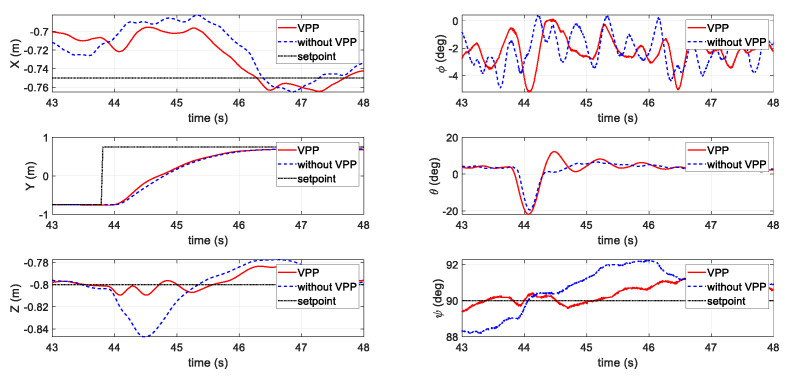
Results of the second flight test scenario with a step command with and without VPP.

**Figure 15 sensors-20-05651-f015:**
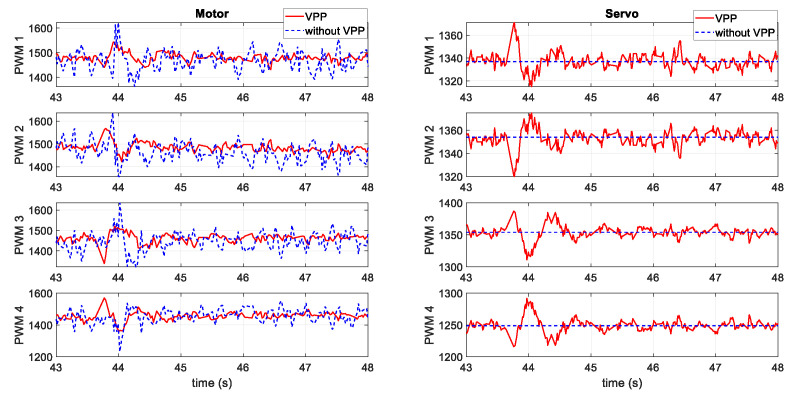
Pulse-width modulation (PWM) signals from the controller to the four motors and four servos.

**Figure 16 sensors-20-05651-f016:**
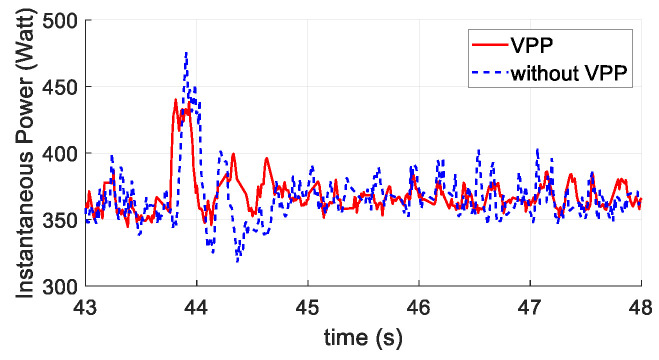
Instantaneous power during the second flight test scenario.

**Figure 17 sensors-20-05651-f017:**
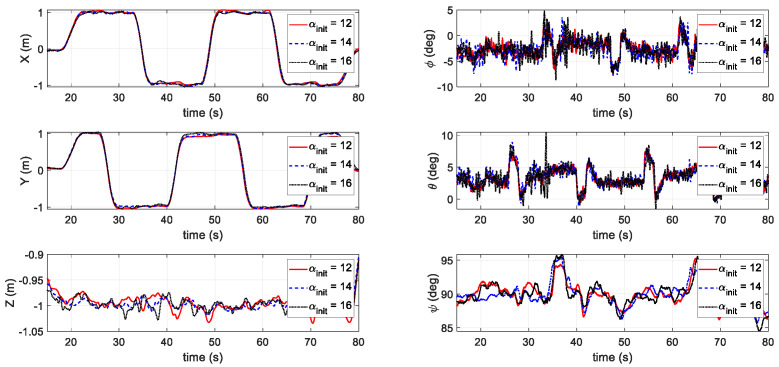
Waypoint mission tracking results for the third flight test scenario.

**Figure 18 sensors-20-05651-f018:**
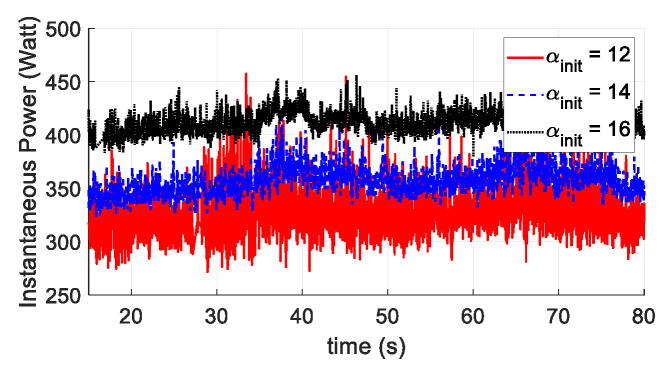
Instantaneous power in waypoint mission tracking for $\alpha_{init}=12{}^{\circ},\;14{}^{\circ},\;\mathrm{and}\;16{}^{\circ}$.

**Table 1 sensors-20-05651-t001:** Symbols used in propeller modeling.

Abbreviations	
V	Airspeed toward the rotational plane (m/s)
u	Propeller’s forward speed =Ω r (m/s)
$V_{r}$	Resultant airspeed toward airfoil (m/s)
θ	Pitch angle measured from the rotational plane (degrees)
α	Angle of attack of the airfoil (degrees)
ϕ	Inflow angle =θ−α (degrees)

**Table 2 sensors-20-05651-t002:** Yaw root-mean-square error (RMSE) of the three simulations.

RMSE of *ψ* (deg)	Without VPP	With VPP
1^st^ scenario	3.610	3.363
2^nd^ scenario	0.128	0.053
3^rd^ scenario	0.128	0.028

**Table 3 sensors-20-05651-t003:** Total energy consumption with and without a VPP controller.

Energy Consumption (W·h)	Without VPP	With VPP
1^st^ flight test scenario	12.49	11.96
2^nd^ flight test scenario	10.08	9.98

**Table 4 sensors-20-05651-t004:** Total energy consumption for $\alpha_{init}=12{}^{\circ},\;14{}^{\circ},\;\mathrm{and}\;16{}^{\circ}$.

	$\alpha_{init}=12{}^{\circ}$	$\alpha_{init}=14{}^{\circ}$	$\alpha_{init}=16{}^{\circ}$
Energy consumption (W·h)	7.85	8.32	8.49

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
