# Peer review of "An Actuator Allocation Method for a Variable-Pitch Propeller System of Quadrotor-Based UAVs"

_sensors, 2020, doi:10.3390/s20195651_

Round 1

Reviewer 1 Report

The paper is well organised, well written and considers an interesting problem, with a good set of practical and simulation results and generally sound mathematical modelling.

There is indeed an improvement in performance that is sufficient to make this proposed system worthy of consideration in smaller UAV.

The main issue with the current version of the paper is that:

  • control inputs and power consumption characteristics are discussed (figure 4, power is  simulated in fig 13, fig 15 and table 3 summarises power consumption), the VPP control inputs that generated the simulation results should have been shown to provide evidence supporting the performance metrics on power consumption. More precisely, plots should be provided for the values of the variable pitch of each propeller and of the prop speeds for at least one of the scenarios.
  • Half of the conclusion is currently focused on future work. I would recommend to focus the conclusion on the lessons learnt from the analysis.

Reviewer 2 Report

This paper introduces a control allocation method for enhancing the attitude following performance and the energy efficiency of a variable-pitch propeller (VPP) system on quadrotor based unmanned aerial vehicles. This paper is a good attempt, but there are some changes that need to be made for the paper to be published.

1. Line 94. You say that 'Despite the previous works regarding the application of VPP systems to small UAVs, no effords have been made to increase the performance of the yaw stability and energy consumption of quadrotor-based UAVs. ' You should explain the main differences of the application of VPP to small UAVs and quadrotor-based UAVs. In other words, you should stress the challenge and difficulty of the application of VPP to quadrotor-based UAVs. So that you can show the contribution and importance of your work.

2. Figure 4(d) is not complete. You lose a ')'.

3. Line 223-235. I am not sure if the notations like 'position_sp' and so on are clerical error or not... Do you want to use subscript or just underline?

4. Figures 7,8,12,14 and 16 are not clear enough. Figure 9 is better. The clarity of these figures are different.

5. Line 383. A clerical error of 'using the the BEM theory'.

Reviewer 3 Report

The authors present a VPP model for enhancing the attitude following performance and the energy efficiency of a variable-pitch propeller (VPP) system on quadrotor based unmanned aerial vehicles. This model was based on a control allocation method using as actuator the VPP allocation method. The flight test results showed that the VPP system can improve the energy consumption during hovering. Although the novelty is not very high, the quality of the writing is. However, the scientific soundness must be improved.

1) To emphasize the minituarization of MEMS sensors, reference is made to them in the introduction. However, this first paragraph does not include any reference. Some like these could be included:

  •  Castaño, F. et al.; Conductance sensing for monitoring micromechanical machining of conductive materials; (2015) Sensors and Actuators, A: Physical, 232, pp. 163-171. DOI: 10.1016/j.sna.2015.05.015
  • A. Merheb, H. Noura and F. Bateman, "Emergency Control of AR Drone Quadrotor UAV Suffering a Total Loss of One Rotor," in IEEE/ASME Transactions on Mechatronics, vol. 22, no. 2, pp. 961-971, April 2017, doi: 10.1109/TMECH.2017.2652399.

2) Usually, the last paragraph of the introduction includes a short introduction to the organization of the article by sections. It is important to introduce this paragraph clearly. 

3) On the other hand, also in the introduction there should be a paragraph dedicated to contributions. These contributions must make it clear that it is proposed to solve the problems and also what differences it has on the revised state of the art.

4) I have found the expression "we" in a couple of sentences. This should not be used in a scientific document. Use the passive voice if possible

5) The conclusions should not be just a mere summary of the work. It is necessary to include those extracted as a discussion of the results in a summarized way and what has finally been achieved with respect to what was initially proposed.

Round 2

Reviewer 3 Report

The authors have modified the paper according to the reviewer´s comment. For this, the quality of presentation and the scientific soundness have improved.

Author Response

Thank you for your kind advice on our paper. We have done English and spell-check.